# Super Enhancer-Regulated *LINC00094* (*SERLOC*) Upregulates the Expression of MMP-1 and MMP-13 and Promotes Invasion of Cutaneous Squamous Cell Carcinoma

**DOI:** 10.3390/cancers14163980

**Published:** 2022-08-17

**Authors:** Minna Piipponen, Pilvi Riihilä, Jaakko S. Knuutila, Markku Kallajoki, Veli-Matti Kähäri, Liisa Nissinen

**Affiliations:** 1Department of Dermatology, University of Turku and Turku University Hospital, Hämeentie 11 TE6, FI-20520 Turku, Finland; 2FICAN West Cancer Centre Research Laboratory, University of Turku and Turku University Hospital, Kiinamyllynkatu 10, FI-20520 Turku, Finland; 3Center for Molecular Medicine, Department of Medicine Solna, Dermatology and Venereology Division, Karolinska Institute, 17176 Stockholm, Sweden; 4Department of Pathology, University of Turku and Turku University Hospital, Kiinamyllynkatu 10, FI-20520 Turku, Finland

**Keywords:** skin cancer, long non-coding RNA, squamous cell carcinoma, super enhancer, matrix metalloproteinase, metastasis

## Abstract

**Simple Summary:**

Cutaneous squamous cell carcinoma (cSCC) is the most common metastatic skin cancer, and its incidence is increasing worldwide. The prognosis of the metastatic disease is poor, and there are no established biomarkers for the assessment of metastasis risk or specific therapeutic targets for advanced or metastatic cSCCs. The role of long non-coding RNAs (lncRNAs) in the progression of cSCC has recently been emphasized. Super enhancers (SE) have been shown to play a role in tumorigenesis and regulate the expression of specific lncRNAs. In this study, we evaluated the role of SE-regulated *BRD3OS* (lncRNA *LINC00094*) in the progression of cSCC. Based on the results, we named this lncRNA *SERLOC*. The results identify *SERLOC* as a biomarker for invasion and metastasis of cSCC and as a putative therapeutic target in advanced cSCC.

**Abstract:**

Long non-coding RNAs (lncRNAs) have emerged as important regulators of cancer progression. Super enhancers (SE) play a role in tumorigenesis and regulate the expression of specific lncRNAs. We examined the role of *BRD3OS*, also named *LINC00094*, in cutaneous squamous cell carcinoma (cSCC). Elevated *BRD3OS* (*LINC00094*) expression was detected in cSCC cells, and expression was downregulated by SE inhibitors THZ1 and JQ1 and via the MEK1/ERK1/2 pathway. Increased expression of *BRD3OS* (*LINC00094*) was noted in tumor cells in cSCCs and their metastases compared to normal skin, actinic keratoses, and cSCCs in situ. Higher *BRD3OS* (*LINC00094*) expression was noted in metastatic cSCCs than in non-metastatic cSCCs. RNA-seq analysis after *BRD3OS* (*LINC00094*) knockdown revealed significantly regulated GO terms *Cell-matrix adhesion*, *Basement membrane*, *Metalloendopeptidase activity*, and KEGG pathway *Extracellular matrix–receptor interaction.* Among the top-regulated genes were *MMP1*, *MMP10*, and *MMP13*. Knockdown of *BRD3OS* (*LINC00094*) resulted in decreased production of MMP-1 and MMP-13 by cSCC cells, suppressed invasion of cSCC cells through collagen I, and growth of human cSCC xenografts in vivo. Based on these observations, *BRD3OS* (*LINC00094*) was named *SERLOC* (super enhancer and ERK1/2-Regulated Long Intergenic non-protein coding transcript Overexpressed in Carcinomas). These results reveal the role of *SERLOC* in cSCC invasion and identify it as a potential therapeutic target in advanced cSCC.

## 1. Introduction

Non-protein coding RNAs (ncRNAs) have been identified as important cellular regulators both in physiological processes, such as development and differentiation and in pathological conditions including cancer [1]. LncRNAs are single-stranded RNA transcripts longer than 200 nucleotides that may interact with DNA, proteins, or other RNAs [2]. They are temporally and tissue-specifically expressed, and this makes lncRNAs versatile regulators in all cellular compartments [3]. Several lncRNAs have been shown to play a role in cancer progression, and they may exert either tumor-suppressive or tumor-promoting functions [4].

Keratinocyte-derived cutaneous squamous cell carcinoma (cSCC) is the most common metastatic skin cancer, and its incidence is increasing worldwide [5]. The metastasis rate of cSCC is 1–4%, and it is responsible for at least 20% of all skin cancer-related mortality [6,7]. Important risk factors of cSCC are solar UV radiation, chronic ulcers, and immunosuppression [8]. During keratinocyte carcinogenesis tumor protein 53 (*TP53*) gene is mutated early in cSCC development, resulting in a marked accumulation of additional UV-induced mutations [9,10,11]. Other genes commonly mutated in cSCC are *NOTCH1*, *HRAS*, and *CDKN2A* [10,12]. In addition, alterations in the tumor microenvironment are crucial for the initiation and development of cSCC [13,14]. The role of lncRNAs in the progression of cSCC is under intense investigation [15]. We have previously identified and characterized two lncRNAs overexpressed in cSCC, p38 inhibited cutaneous squamous cell carcinoma-associated lincRNA (*PICSAR*) [16,17] and p53 regulated carcinoma-associated STAT3-activating long intergenic non-protein coding transcript (*PRECSIT*) [18]. *PICSAR* promotes cSCC growth by increasing ERK1/2 activity via downregulation of *DUSP6* [16], decreases adhesion, and promotes migration of cSCC cells by downregulating the expression of α2β1 and α5β1 integrins [17]. *PRECSIT* increases cSCC cell invasion by regulating MMP expression via STAT3 signaling [18]. However, the role of lncRNAs in cSCC progression is still largely unknown.

In this study, we demonstrate that *BRD3OS* (BRD3 opposite strand), also named lncRNA LINC00094, is specifically overexpressed by cSCC cells in culture and in vivo. The expression level of *BRD3OS* (*LINC00094*) is markedly upregulated in tumor cells in UV-induced primary cSCCs and in their metastases compared to normal skin, and precancerous forms of actinic keratosis (AK) and cSCC in situ (cSCCIS). Furthermore, *BRD3OS* (*LINC00094*) expression level is stronger in metastatic cSCCs than in non-metastatic cSCCs. The expression of *BRD3OS* (*LINC00094*) in cSCC cells is downregulated by super enhancer (SE) inhibitors THZ1 and JQ1. *BRD3OS* (*LINC00094*) regulates the expression of matrix metalloproteinases (MMPs) MMP-1 and MMP-13 and promotes the invasion of cSCC cells. Based on these observations, *BRD3OS* (*LINC00094*) was named *SERLOC* (Super enhancer and ERK1/2 Regulated Long Intergenic non-protein coding transcript Overexpressed in Carcinomas). Altogether these results identify *SERLOC* as a super enhancer-regulated lncRNA, which promotes invasion of cSCC by regulating MMP expression.

## 2. Materials and Methods

### 2.1. Cell Culture

Normal human epidermal keratinocytes (NHEKs) were isolated from the skin of healthy individuals undergoing mammoplasty [19], and NHEK-PC was from PromoCell (Heidelberg, Germany). Primary non-metastatic (UT-SCC12A, UT-SCC91, UT-SCC105, UT-SCC111, and UT-SCC118) and metastatic (UT-SCC7, UT-SCC59A, and UT-SCC115) cSCC cell lines were established from surgically removed cSCCs in Turku University Hospital [19]. The authenticity of all cSCC cell lines has been verified by short tandem repeat profiling [20]. The spontaneously immortalized human keratinocyte line (HaCaT) lacking functional p53 and three HaCaT-derived Ha-ras–transformed cell lines (A5, II-4, and RT3), which represent in vitro models for progressive stages of cSCC tumors [21], was kindly provided by Dr. Norbert E. Fusenig (The German Cancer Research Center, Heidelberg, Germany). A5 is a benign tumorigenic HaCaT cell line, II4 forms invasive tumors, and RT3 forms metastatic SCC [21]. Cells were cultured as previously described [22]. Super-enhancer inhibitors THZ1 (inhibitor of CDK7) and JQ1 (inhibitor of BRD4) were used. To determine *BRD3OS* (*LINC00094*) expression after inhibition of CDK7 and BRD4, cSCC were treated with THZ1 (100 nM) and JQ1 (5 µM) for 24 or 48 h (MedChemExpress, Monmouth Junction, NJ, USA). For inhibition of MEK1/2, cSCC cells were treated with PD98059 (30 μM) for 24 h (Calbiochem, La Jolla, CA, USA).

### 2.2. Real-Time Quantitative PCR

Total RNA was extracted from cultured NHEKs and cSCC cells using an RNeasy mini kit (Qiagen, Germantown, MD, USA), and 1 µg of total RNA was reverse transcribed into cDNA with random hexamer and M-MLV Reverse Transcriptase H Minus (both from Promega, Madison, WI, USA) for real-time quantitative reverse transcriptase-PCR (qRT-PCR) analysis. Specific primers and probes for *BRD3OS* (*LINC00094*) were designed with RealTimeDesign Software (https://www.biosearchtech.com; last accessed 17 April 2012; LGC Biosearch Technologies, Teddington, UK) and purchased from Oligomer (Helsinki, Finland) (Appendix A). Primers and probes for *MMP1*, *MMP10*, and *MMP13* (Appendix A) were designed as previously described [23]. qRT-PCR reactions were performed utilizing the QuantStudio 12K Flex (Thermo Fisher Scientific) at the Finnish Functional Genomics Centre in Turku, Finland. qRT-PCR amplification was done using the following protocol: hold stage 2 min at 50 °C, 10 min at 95 °C, and PCR stage for 40 cycles 0.15 min at 95 °C and 1 min at 60 °C. *ACTB* (β-actin) or *GAPDH* mRNA was used as reference (Appendix A). Samples were analyzed using the standard curve method in three parallel reactions with threshold cycle values <5% of the mean threshold cycle.

### 2.3. Tissue Samples

Tissue microarrays (TMAs) consisting of samples from normal sun-protected and sun-exposed skin (*n* = 24), AK (*n* = 67), cSCCIS (*n* = 60), non-metastatic invasive cSCC (*n* = 119), metastatic cSCC (*n* = 76), and cSCC metastases (*n* = 8) were generated from the archival paraffin blocks from the Department of Pathology, Turku University Hospital and Auria Biobank, Turku University Hospital and University of Turku [24]. TMAs were generated from formalin-fixed paraffin-embedded tissue samples as described earlier [25].

### 2.4. RNA In Situ Hybridization

TMAs were subjected to RNA in situ hybridization (RNA-ISH) with a specific probe for *BRD3OS* (*LINC00094*) (Hs-BRD3OS-O1, #823849) (Advanced Cell Diagnostics (ACD), Newark, CA, USA) and analyzed with RNAscope ISH Assay (ACD, Newark, CA) by Bioneer A/S (Hørsholm, Denmark). An automated Ventana Discovery Ultra slide-staining system (Roche) was used to accomplish the ISH assay as previously described [18]. Specific mRNAs of bacterial DapB (4-hydroxy-tetrahydrodipicolinate reductase) and human PPIB (Cyclophilin B) were used as negative and positive controls, respectively (both from ACD) as previously described [18]. The expression of *BRD3OS* (*LINC00094*) was illustrated with Zeiss Axioscan (Carl Zeiss AG, Oberkochen, Germany) at 20× magnification and visualized with Zen lite (Carl Zeiss Microscopy, München, Germany) or a Pannoramic 1000 Slide Scanner (3DHistech, Budapest, Hungary). The visible cytoplasmic and nuclear particles for *BRD3OS* (*LINC00094*) transcripts were counted, and the tissue samples were classified on the basis of the detected particles in one cell and the distribution of particles in TMAs by three independent observers (P.R., L.N., and M.K.). *BRD3OS* (*LINC00094*) was classified as negative (−) when single particles were noted in single cells, weak positive (+) when single particles were noted in several cells, moderate positive (++) when two particles were detected in several cells, and strong positive (+++) when more than 2 particles were detected widely in cells.

### 2.5. Adenoviral Infection

HaCaT cells were infected with Escherichia coli β-galactosidase containing control (RAdLacZ) and constitutively active MEK1-containing (RAdMEK1ca) adenoviral vectors and incubated in a conditioned medium containing 0.5% fetal calf serum at MOI 600 for 6 h. After 48 h, cell lysates were collected and subjected to qPCR analysis. RAdMEK1ca and RAdLacZ adenoviral vectors were kindly provided by Dr. Marco Foschi (University of Florence, Italy) [26] and Dr. Gavin W.G. Wilkinson (University of Cardiff, UK) [27]. To inhibit activation of MEK1, cells were serum starved for 24 h and then treated with small molecule inhibitor PD98059 (30 μM, Calbiochem, La Jolla, CA, USA).

### 2.6. RNA-Sequencing

cSCC cells were cultured to 50% confluence and transfected with negative control or *BRD3OS* siRNA4 (75 nM). RNA was isolated from transfected cells (*n* = 3; UT-SCC7, UT-SCC12A, and UT-SCC59A) 72 h later using miRNeasy Mini Kit (Qiagen). For RNA-sequencing (RNA-seq) the Illumina HiSeq3000 was used (Illumina, San Diego, CA, USA) at the Finnish Functional Genomics Centre, Turku. For library preparation, Illumina TruSeq Stranded total RNA Sample Preparation Kit was used. The reads obtained from the instrument were base called using the instrument manufacturer’s Bcl2fastq2 version 2.17 base-calling software. The reads were aligned against the human reference genome (hg38), and for data normalization the TMM (trimmed mean of M values) method was used (R version 3.3/Bioconductor package edgeR version 3.331, R Foundation for Statistical Computing, Vienna, Austria) [28]. R package Limma version 3.1032 and *t*-test were used for statistical analysis of the mean expression values between control siRNA and *BRD3OS* siRNA treated cSCC cells. For filtering the results, fold change (FC) log 1.0 and *p* < 0.01 were used. Morpheus (https://software.broadinstitute.org/morpheus (last accessed 30 June 2022)) online tool was used to generate heatmaps. RNA-seq data were further analyzed using the Gene Ontology Enrichment Analysis (Gene Ontology, http://www.geneontology.org (last accessed 31 October 2018)) and the Kyoto Encyclopedia of Genes and Genomes Pathway Analysis (KEGG, http://www.genome.jp/kegg (last accessed 31 October 2018)). RNA-seq data of *BRD3OS* (*LINC00094*) knockdown cSCC cells are available online at the Gene Expression Omnibus (accession number GSE205981).

### 2.7. Western Blot Analysis

cSCC cells were cultured to 50% confluence and transfected with commercially available siRNAs (75 nM) targeting *BRD3OS* (*LINC00094*), MMP-1, or MMP-13 (Qiagen, Hilden, Germany) by using siLentFect lipid reagent (Biorad, Hercules, CA, USA) (Appendix A). MMP levels in a serum-free medium of cSCC cell cultures were determined by Western blot analysis 72 h after siRNA transfection using antibodies specific for MMP-1 (1:1000, MAB3307, 41-1E5; Merck Millipore, Temecula, CA, USA) and MMP-13 (1:500, MAB3321; Merck Millipore). Expression of tissue inhibitor of matrix metalloproteinase 1, TIMP-1 (1:1000, MAB3300; Merck Millipore), was used as a control. Protein expression was quantitated using IRDye labeled secondary antibodies and the LI-COR Odyssey CLx fluorescent imaging system (LI-COR Biosciences, Lincoln, NE, USA). Quantitation was prepared by using Image Studio software (LI-COR Biosciences).

### 2.8. Human cSCC Xenografts

cSCC cells (UT-SCC7) were transfected with *BRD3OS* siRNA or control siRNA and incubated for 72 h. Cells (5 × 10^6^) were injected subcutaneously in 100 μL volume in phosphate-buffered saline into the backs of 6-week-old female severe combined immunodeficient (SCID) mice (CB17/Icr-Prkdc^scid^/IcrIcoCrl) (Charles River Laboratories, Wilmington, MA) (control siRNA, *n* = 7; *BRD3OS* siRNA, *n* = 8). Tumor size was measured twice a week, and the tumor volume was calculated as follows: V = (length × width^2^)/2 [18]. Mice were sacrificed 16 days after tumor implantation.

### 2.9. Invasion Assays

To study cell invasion in culture, cSCC cells were transfected with negative control and *BRD3OS* (*LINC00094*), MMP-1, or MMP-13 targeting siRNAs (75 nM) and cultured for 24 h. Transfected cells were plated on a collagen type I–coated (5 μg/cm^2^, PureCol; Advanced BioMatrix, San Diego, CA, USA) ImageLock 96-well plate (Essen Bioscience, Ann Arbor, MI, USA), and cells were allowed to adhere overnight. The cell monolayer was scratched using an Incucyte wound maker (Essen Bioscience), and collagen type I solution was added by mixing type I collagen (PureCol) with 5× Dulbecco’s Modified Eagle Medium and 0.2 mol/L HEPES buffer (pH 7.4) at a ratio of 7:2:1, respectively. Finally, 1 mol/L NaOH was added to obtain a final pH of 7.4. Collagen was allowed to polymerize for 2 h at 37 °C, and a cell culture medium with 0.5% fetal calf serum was added on top. The gap closure was imaged using the IncuCyte S3 real-time cell imaging system (Essen Bioscience), and the relative cell invasion was quantitated using the IncuCyte S3 software version 2020A (Essen Bioscience).

### 2.10. Statistical Analysis

Statistical analyses were performed using GraphPad Prism, version 9.1.0 (GraphPad Software, San Diego, CA, USA). The sample size was determined to be adequate for the statistical analysis of the data. To determine the significance of differences between two or more sample groups, a two-tailed Student’s *t*-test and two-way ANOVA were used. Two-tailed X^2^ test or Fisher’s exact test was used for statistical comparisons of the RNA-ISH analysis of tissue samples. Results were considered statistically significant when bidirectional *p*-values were <0.05 (* <0.05, ** <0.01, and *** <0.001).

## 3. Results

### 3.1. BRD3OS (LINC00094) Is Overexpressed in cSCC Cells

We have previously shown differential expression of several lncRNAs in cSCC cells compared with NHEKs [16]. *BRD3OS* (*LINC00094*) was identified as one of the upregulated lncRNAs in cSCC cells compared to NHEKs based on RNA-seq [16]. *BRD3OS* (*LINC00094*) is transcribed from the opposite strand of bromodomain containing 3 gene (*BRD3*) (Figure 1A). Upregulation of *BRD3OS* (*LINC00094*) was detected in cSCC cell lines (*n* = 8) compared with NHEKs (*n* = 4) by qRT-PCR (Figure 1B). In accordance with this, we also noticed increased *BRD3* expression in cSCC cells compared to NHEKs by qRT-PCR (Appendix A). According to our RNA-seq data, the expression of *BRD3OS* (*LINC00094*) and *BRD3* showed a positive correlation (Appendix A). Based on mRNA expression data from the GEPIA database [29], *BRD3OS* (*LINC00094*) and *BRD3* were also upregulated in head and neck SCCs (HNSCCs), and a positive correlation was noted in their expression (Appendix A). The analysis of *BRD3OS* (*LINC00094*) expression with RNA-ISH revealed a specific, mainly cytoplasmic signal for this lncRNA in cSCC cells (Figure 1C). Furthermore, analysis of tissue sections of xenografts established with human cSCC cells (UT-SCC7) with RNA in situ hybridization (RNA-ISH) revealed specific expression of *BRD3OS* (*LINC00094*) in tumor cells (Figure 1D).

### 3.2. BRD3OS (LINC00094) Is Regulated by Super Enhancer in cSCC Cells

*BRD3OS* (*LINC00094*) expression has been previously shown to be regulated by a super enhancer (SE) in esophageal SCC [30]. Based on the SE-archive version 3.0 online (SEA v. 3.0, http://sea.edbc.org, (last accessed 31 May 2022) Computational Biology Research Center, Harbin, China) [31], we found a SE near the *BRD3OS* gene (Figure 1A). We treated cSCC cells with THZ1, a covalent inhibitor of CDK7, and JQ1, a specific inhibitor of BRD4, which both have been shown to selectively target SE-driven transcriptional programs in cancer [32,33]. Following treatment with THZ1 or JQ1, *BRD3OS* (*LINC00094*) expression was significantly decreased in cSCC cells (Figure 1E), indicating that the expression of this *lncRNA* is driven by a super enhancer.

### 3.3. BRD3OS (LINC00094) Is Expressed by cSCC Tumor Cells In Vivo

Paraffin-embedded formalin-fixed TMA samples containing normal skin (*n* = 24), premalignant AK (*n* = 67), precancerous cSCCIS (*n* = 60), UV-induced primary non-metastatic invasive cSCC (*n* = 119), primary metastatic cSCC (*n* = 76) and cSCC metastases (*n* = 8) were analyzed with RNA-ISH for the expression of *BRD3OS* (*LINC00094*) in cSCC carcinogenesis in vivo. Particles of *BRD3OS* (*LINC00094*) were noted mainly in the cell cytoplasm and less abundantly in nuclei. The expression of *BRD3OS* (*LINC00094*) was strong in cSCCs and cSCC metastases (Figure 2A–C), whereas in AKs (Figure 2E) and cSCCISs (Figure 2F), the expression was weaker. In normal skin, the expression was weaker than in AK and cSCCIS (Figure 2D), The *BRD3OS* (*LINC00094*) positive particles in the cytoplasm and nucleus of epidermal and tumor cells were counted, and the tissue samples were classified based on the number of particles in one cell. In addition, the distribution of particles in tissue samples was estimated and taken into account in the classification (Figure 2A–G).

*BRD3OS* (*LINC00094*) expression was scored as negative (−) when single particles were detected in single cells, weak positive (+) when single particles were noted in several cells, moderate positive (++) when two particles were detected in several cells and strong positive (+++) when more than two particles were detected in cell cytoplasm or nuclei scattered widely in samples. The analysis showed that there was no strong (more than two particles in cell) (+++) expression of *BRD3OS* (*LINC00094*) in normal skin samples (0% of cases), and the strong (+++) positive expression was noted significantly more in premalignant lesions (AK and cSCCIS) (42% of cases), invasive primary non-metastatic cSCCs (55%), primary metastatic cSCCs (70%), and cSCC metastases (75%) than in normal skin (0%) (Figure 2G). In addition, there were significantly less *BRD3OS* (*LINC00094*) positivity in AKs and cSCCISs compared to non-metastatic cSCCs and metastatic cSCCs but more than in normal skin (Figure 2G). Furthermore, significantly stronger expression was detected in metastatic cSCCs compared to non-metastatic cSCCs (Figure 2G). In addition, the expression of *BRD3OS* (*LINC00094*) in cSCC metastases was as strong as in primary metastatic cSCCs (Figure 2G). For positive and negative controls, human PPIB (Cyclophilin B) and bacterial DapB (4-hydroxy-tetrahydrodipicolinate reductase) mRNAs were used, respectively (Appendix A). The results indicate that *BRD3OS* (*LINC00094*) expression is upregulated during the progression of cSCC to invasive and metastatic stage.

### 3.4. BRD3OS (LINC00094) Expression Is Regulated by ERK1/2 Pathway

Basal activation of ERK1/2 and p38 MAPKs is detected in cSCC cells in culture and in vivo [34,35,36]. To investigate the significance of *BRD3OS* (*LINC00094*) in epidermal carcinogenesis, expression was studied in the HaCaT cell line, which is derived from human epidermal keratinocytes and lacks functional p53, as well as in three HaCaT-derived Ha-ras–transformed cell lines (A5, II-4, and RT3), which are representative in vitro models for progressive stages of cSCC tumors. Malignant II-4 and RT3 cells also show more potent ERK1/2 activation compared to benign A5 and HaCaT cells [36,37].

*BRD3OS* (*LINC00094*) levels were highest in the nontumorigenic HaCaT cells and gradually lower compared with benign Ha-ras–transformed A5, II-4, and RT-3 cells, respectively (Figure 3A). Treatment with PD98059, a specific inhibitor of MEK1 activation and the MAP kinase cascade, led to a significant increase in *BRD3OS* (*LINC00094*) expression in HaCaT cells (Figure 3B). In addition, HaCaT cells were transduced with recombinant adenoviruses containing constitutively active MEK1 (RAdMEK1ca) or β-galactosidase (RAdLacZ, control vector). In accordance with the observed decrease in *BRD3OS* (*LINC00094*) expression in the Ha-ras–transformed HaCaT cell lines (Figure 3A), *BRD3OS* (*LINC00094*) expression was significantly decreased in HaCaT cells transduced with RAdMEK1ca (Figure 3C). Furthermore, treatment with PD98059 in RAdLacZ and RAdMEK1ca transduced HaCaT cells led to significantly increased expression of *BRD3OS* (*LINC00094*) (Figure 3C). Together, these results suggest that *BRD3OS* (*LINC00094*) expression is downregulated by the ERK1/2 pathway.

### 3.5. Knockdown of BRD3OS (LINC00094) Inhibits the Expression of MMP-1 and MMP-13

To study the functional role of *BRD3OS* (*LINC00094*) in cSCC in more detail, we performed an RNA-seq analysis of cSCC cells after *BRD3OS* siRNA knockdown (Figure 4A). Gene enrichment analysis revealed several gene ontology (GO) terms related to cell adhesion and extracellular space, such as *Cell-matrix adhesion*, *Collagen fibril organization*, *Integrin binding*, and *Metalloendopeptidase activity* (Figure 4B). Several matrix metalloproteinases (MMPs), including *MMP10*, *MMP13*, and *MMP7*, were among the top downregulated genes in *BRD3OS* (*LINC00094*) knockdown cSCC cells (Figure 4C), and several other MMP-gene family members clearly stand out downregulated within the GO *Metalloendopeptidase activity* term (Figure 4D). *MMP1*, *MMP10*, and *MMP13* were among the significantly regulated MMP genes after *BRD3OS* (*LINC00094*) knockdown (Appendix A). Decreased expression of *MMP1*, *MMP10*, and *MMP13* was further validated in cSCC cells after *BRD3OS* (*LINC00094*) knockdown by qRT-PCR and Western blotting (Figure 5 and Appendix A).

### 3.6. Knockdown of BRD3OS (LINC00094) Inhibits Invasion of cSCC Cells by Downregulating MMP-1 and MMP-13 Production

The RNA-seq analysis showed, that *BRD3OS* (*LINC00094*) siRNA knockdown resulted in decreased expression of several MMP genes shown to degrade various ECM components, which lead us to investigate the role of *BRD3OS* (*LINC00094*) in cSCC cell invasion. Invasion of *BRD3OS* (*LINC00094*) knockdown cells through type I collagen matrix was significantly decreased compared to negative control siRNA treated cSCC cells (Figure 6A–C). Knockdown of MMP-1 or MMP-13 (Appendix A) inhibited the invasion of control siRNA—transfected cells through collagen I but had no effect on cell invasion after *BRD3OS* (*LINC00094*) knockdown (Figure 6C). Lastly, cSCC cells were transfected with *BRD3OS* siRNA and negative control siRNA and injected subcutaneously into the back of the SCID mice for studying the role of *BRD3OS* (*LINC00094*) in tumor growth in vivo. Xenograft growth was measured twice a week, and tumors were harvested 16 days after implantation. *BRD3OS* (*LINC00094*) knockdown decreased tumor growth compared with control tumors (Figure 6D). *BRD3OS* (*LINC00094*) expression was specifically detected at the invasive edge of the xenograft tumors (Figure 1B), supporting our findings above of *BRD3OS* (*LINC00094*) in regulation of MMP expression and invasion of cSCC cells.

## 4. Discussion

LncRNAs are still a largely uncharacterized group of non-coding RNAs with diverse regulatory roles in various biological processes. LncRNAs are strictly regulated, and they show cell and tissue-specific expression and subcellular localization [3]. These issues make lncRNAs interesting potential biomarkers and therapeutic targets in cancer. The role of lncRNAs in the pathogenesis of keratinocyte-derived skin cancers is not well known, but there is growing evidence for the role of these RNA molecules in the progression of cSCC [15]. Previously, we have shown the function of lncRNAs *PICSAR* and *PRECSIT* in the progression of cSCC [16,17,18]. *PICSAR* regulates the proliferation of cSCC cells by increasing ERK1/2 activity [16]. *PICSAR* also regulates the migration of these cells by downregulating α2β1 and α5β1 integrins [17]. On the other hand, *PRECSIT* has been shown to increase the invasion of cSCC cells by regulating the expression of MMPs via STAT3 signaling [18]. Additionally, recent studies have implicated lncRNAs *HOTAIR* [38], *NEAT1* [39], and *MALAT1* [40] in the progression of cSCC. Furthermore, *LINC00319* has been noted to regulate the invasion of cSCC cells [41].

The whole transcriptome expression analysis identified *BRD3OS* (*LINC00094*) as one of the significantly upregulated lncRNAs in cSCC cells compared with NHEKs [16]. In this study, overexpression of *BRD3OS* (*LINC00094*) in primary and metastatic cSCC cell lines compared with NHEKs was confirmed by qRT-PCR. Additionally, the expression of *BRD3OS* (*LINC00094*) was detected in cSCC cells in culture and in cSCC-derived xenograft tumors using RNA-ISH. Based on the ISH of cultured cSCC cells, *BRD3OS* (*LINC00094*) is localized mainly in the cytoplasm and perinuclear space. In previous studies, *BRD3OS* (*LINC00094*) has been shown to be upregulated during cancer progression. *BRD3OS* (*LINC00094*) has been shown to be expressed by esophageal cancer (ESCC) cells and to induce the growth and survival of these cancer cells [30]. On the other hand, *BRD3OS* (*LINC00094*) has been shown to be downregulated in lung cancer compared to normal tissue [42]. *LINC00094* is encoded by the *BRD3OS* gene that is predicted to encode an 84 amino acid protein, and this open reading frame is conserved in many mammals and some amphibians and fish [43]. Approximately 40% of lncRNAs are translated, and cytoplasmic lncRNAs such as *BRD3OS* (*LINC00094*) are more often translated than nuclear lncRNAs [44]. On the other hand, the translatability of lncRNAs does not necessarily indicate whether the translated peptide is biologically functional or stable enough to be detected, and these RNA molecules may have both RNA- and protein-related functions [44].

To study the basal regulation of *BRD3OS* (*LINC00094*) expression, an immortalized nontumorigenic cell line HaCaT, derived from human epidermal keratinocytes and *Ha-ras*-transformed HaCaT cell lines (A5, II-4, and RT3) were used. The results demonstrate that the expression of *BRD3OS* (*LINC00094*) was decreased in Ha-ras transformed HaCaT cell lines. The increased phosphorylation of ERK1/2 has been demonstrated in HaCaT cell lines RT3 and II4 showing the highest phosphorylation of ERK1/2 [36]. Treatment of HaCaT cells with an ERK1/2 inhibitor increased, and infection of HaCaT cells with constitutively active MEK1 adenovirus decreased the expression of *BRD3OS* (*LINC00094*). These results show that the basal expression of *BRD3OS* (*LINC00094*) is decreased by MEK1/ERK1/2 pathway in HaCaT cells. The phosphorylation level of ERK1/2 varies both in cSCC lines in culture and in cSCC tumors in vivo [35,36]. Our results also show that *BRD3OS* (*LINC00094*) is not expressed by all cSCC tumor cells in vivo. Based on these findings, the heterogenicity of cSCC tumors may indicate that in some cSCC cells in which ERK1/2 activity is low, the expression of *BRD3OS* (*LINC00094*) might increase and thus promote the invasion of these cells.

Super enhancers have been shown to present a role in tumorigenesis, indicating that they could be promising therapeutic targets for cancer treatment [42,45]. Targeted small molecule inhibitors (SMI) have been developed to specifically block the interaction between SE regions and their corresponding complexes. The SMIs designed for the treatment of cancer include, for example, BRD4 inhibitor (JQ1) [46] and CDK7 inhibitor (THZ1) [47]. SMIs against CDK7 and BRD4 are interesting drug candidates and have entered clinical trials [42]. In this study, *BRD3OS* (*LINC00094*) was noted to be downregulated by THZ1 and JQ1, demonstrating that it is a super enhancer-regulated lncRNA in cSCC cells.

To investigate the molecular mechanism of *BRD3OS* (*LINC00094*) in more detail, total RNA-seq was performed for cSCC cells after *BRD3OS* (*LINC00094*) knockdown. Knockdown of *BRD3OS* (*LINC00094*) significantly regulated the genes belonging to GO terms and KEGG pathways related to the invasion of cSCC cells. Interestingly, the regulation of several cSCC invasion-associated MMPs was noted after *BRD3OS* (*LINC00094*) knockdown. Notably, the genes encoding for these MMPs are all located in the MMP gene cluster in locus 11q22.3 [48], suggesting that this MMP gene cluster is regulated by super enhancer-regulated *BRD3OS* (*LINC00094*).

MMP-1 and MMP-13 were also shown to be downregulated after *BRD3OS* (*LINC00094*) knockdown at the protein level. Previously, MMP-1 and MMP-13 have been shown to be specifically expressed by tumor cells in cSCC [49,50]. In addition. MMP-1 and MMP-13 have been demonstrated to be important regulators of cSCC cell invasion [51,52,53]. Based on these findings, the role of *BRD3OS* (*LINC00094*) in cSCC cell invasion was investigated. It was noted that knockdown of *BRD3OS* (*LINC00094*) inhibited the invasion of cSCC cells through collagen I by regulating the expression of MMP-1 and MMP-13. It is tempting to suggest that in cSCC cells where ERK1/2 activity is low, *BRD3OS* (*LINC00094*) expression is elevated, and the increased expression of this lncRNA then promotes the invasion of these cancer cells.

## 5. Conclusions

In this study, *BRD3OS* (*LINC00094*) was shown to be upregulated in cSCC cells compared to NHEKs. Additionally, increased expression of *BRD3OS* (*LINC00094*) was noted in tumor cells in cSCC in vivo compared with normal skin, AK, and cSCCIS. The expression of *BRD3OS* (*LINC00094*) in cSCC cells was downregulated by SE inhibitors THZ1 and JQ1 in cSCC cells. Knockdown of *BRD3OS* (*LINC00094*) resulted in significantly decreased invasion of cSCC cells through collagen type I and suppressed the growth of human cSCC xenografts in vivo. Additionally, *BRD3OS* (*LINC00094*) knockdown decreased the expression of invasion-related proteinases MMP-1 and MMP-13. Based on these observations, *BRD3OS* (*LINC00094*) was named *SERLOC* (Super Enhancer and ERK1/2 Regulated Long Intergenic non-protein coding transcript Overexpressed in Carcinomas). These results provide evidence for the role of *SERLOC* in promoting the invasion of cSCC cells by regulating the production of invasion-associated MMPs and suggest SE-regulated *SERLOC* as a biomarker for cSCC metastasis and as a potential therapeutic target in the treatment of locally advanced and metastatic cSCC.

## Figures and Tables

**Figure 1 cancers-14-03980-f001:**
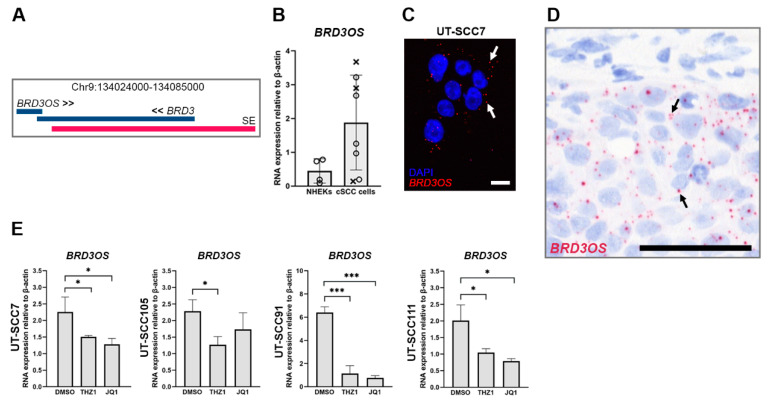
Overexpression of *BRD3OS* (*LINC00094*) in cSCC cells is regulated by super enhancer. (**A**) *BRD3OS* (*LINC00094*) gene is located near a super enhancer (SE, red bar). (**B**) *BRD3OS* (*LINC00094*) expression was measured by quantitative real-time reverse transcriptase PCR (qRT-RCR) in normal human epidermal keratinocytes (NHEKs) (*n* = 4) and cSCC cells (*n* = 5, primary, round dots; *n* = 3 metastatic, crosses). (**C**) Expression and cellular localization of *BRD3OS* (*LINC00094*) (white arrows) were examined by RNA in situ hybridization (RNA-ISH). Scale bar = 20 µm. (**D**) *BRD3OS* (*LINC00094*) expression (black arrows) was determined in cSCC xenograft tumors (UT-SCC7) by RNA-ISH. Scale bar = 50 µm. (**E**): *BRD3OS* (*LINC00094*) expression in cSCC cell lines was determined by qRT-PCR 24 h (UT-SCC7 and −105) or 48 h (UT-SCC91 and −111) after treatment with THZ1 (100 nM) or JQ1 (5 µM). β-Actin mRNA levels were determined as a reference gene. Mean + SD is shown, * *p* < 0.05; *** *p* < 0.001, Student’s *t*-test.

**Figure 2 cancers-14-03980-f002:**
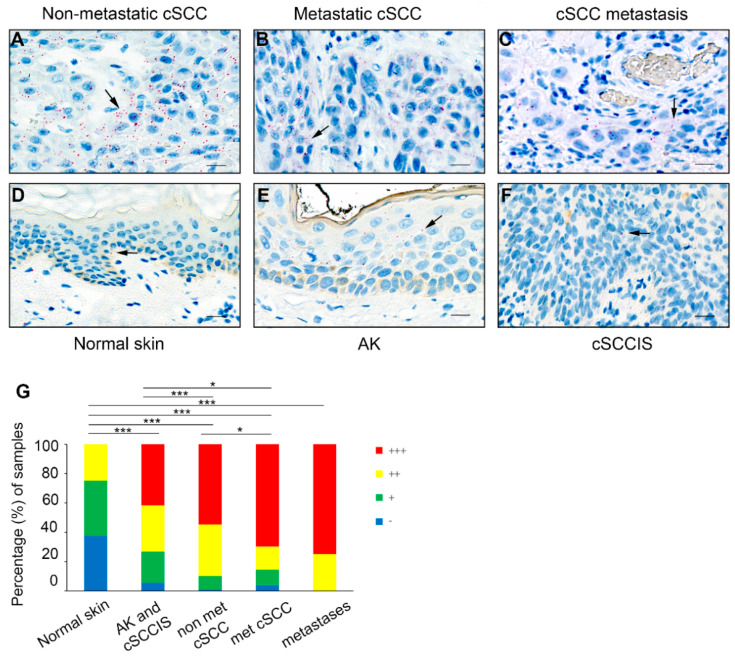
Expression of *BRD3OS* (*LINC00094*) in cSCC tumor cells in vivo. (**A**–**F**) RNA in situ hybridization (RNA-ISH) expression of *BRD3OS* (*LINC00094*) in tissue sections. (**A**) UV-induced primary non-metastatic invasive cSCC (*n* = 119), (**B**) primary metastatic cSCC (*n* = 76), (**C**) cSCC metastasis (*n* = 8), (**D**) normal skin (*n* = 24), (**E**) actinic keratosis (AK; *n* = 67) and (**F**) cSCC in situ (cSCCIS; *n* = 60). In invasive non-metastatic cSCCs (**A**), primary metastatic cSCCs (**B**), and cSCC metastases (**C**), the expression of mainly cytoplasmic *BRD3OS* (*LINC00094*) was stronger than in normal skin (**D**), AKs (**E**) or cSCCISs (**F**). Arrows illustrate the *BRD3OS* (*LINC00094*) positive spots in cells. Scale bar = 20 µm. (**G**) The expression of *BRD3OS* (*LINC00094*) was classified as negative (−), weak (+), moderate (++), and strong (+++) according to the extensity and number of positive spots in cells. * *p* < 0.05, *** *p* < 0.001 by two-tailed X^2^ or Fisher’s exact test.

**Figure 3 cancers-14-03980-f003:**
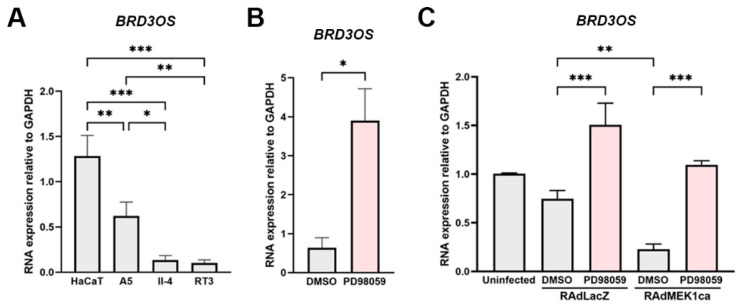
*BRD3OS* (*LINC00094*) expression by cSCC cells is regulated by ERK1/2 pathway. *BRD3OS* (*LINC00094*) expression was determined by qRT-PCR in (**A**) HaCaT cell lines, (**B**) PD98059-treated (30 μM) HaCaT cells, and (**C**) in HaCaT cells expressing constitutively active MEK1 (RAdMEK1ca) or β-galactosidase (RAdLacZ, control vector). GAPDH mRNA levels were determined as a reference gene. Mean + SD is shown, * *p* < 0.05; ** *p* < 0.01; *** *p* < 0.001. Student’s *t*-test.

**Figure 4 cancers-14-03980-f004:**
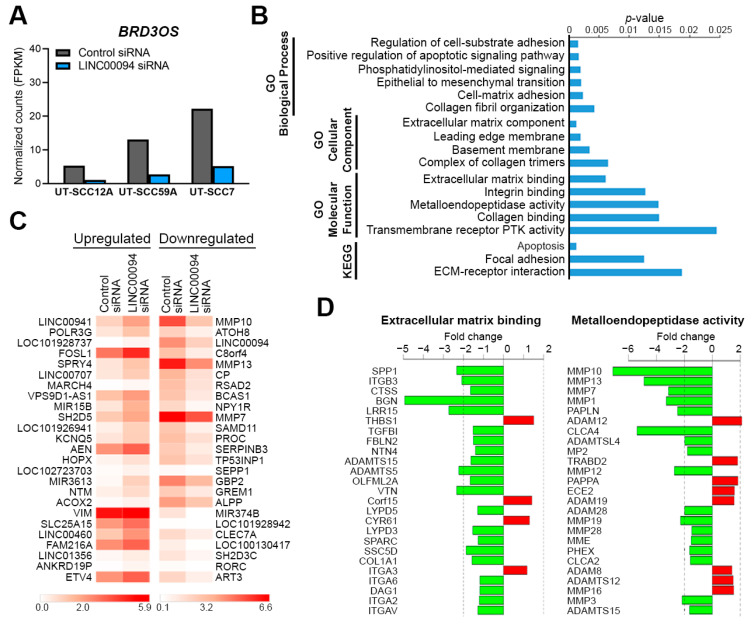
RNA-seq analysis of *BRD3OS* (*LINC00094*) knockdown cSCC cells. cSCC cells were transfected with negative control siRNA or *BRD3OS* siRNA4, and after 3 days, total RNA was isolated, and RNA-seq was prepared. (**A**) *BRD3OS* (*LINC00094*) knockdown efficiency in three cSCC cell lines (UT-SCC12, UT-SCC59A, and UT-SCC7) in the RNA-seq analysis. (**B**) Gene enrichment analysis showing the top Gene Ontology (GO) and Kyoto Encyclopedia of Genes and Genomes (KEGG) terms of differentially expressed genes (full ranked list) in *BRD3OS* (*LINC00094*) knockdown cSCC cells. (**C**) Heatmap showing the top 25 up- and downregulated genes (log2FC ≥ 1 or ≤−1, *p* < 0.05) in *BRD3OS* (*LINC00094*) knockdown cSCC cells. (**D**) Gene plots showing the top 25 ranked genes included in the respective GO terms.

**Figure 5 cancers-14-03980-f005:**
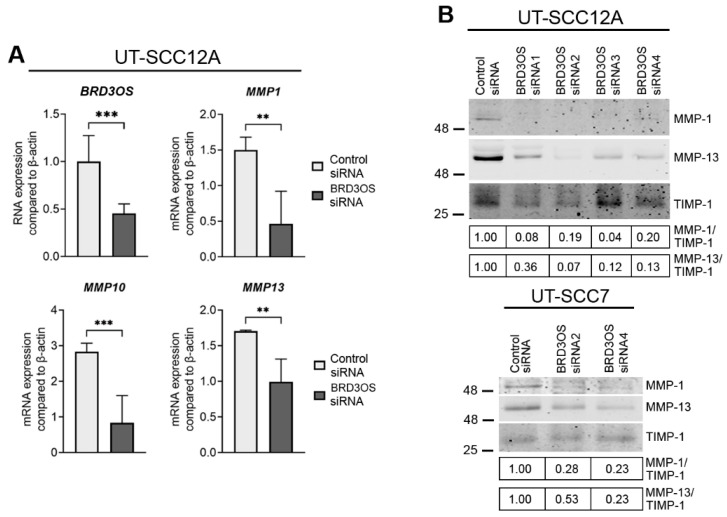
The expression of MMP-1 and MMP-13 is downregulated in cSCC cells after *BRD3OS* (*LINC00094*) knockdown. (**A**) *BRD3OS* (*LINC00094*), *MMP1*, *MMP10*, and *MMP13* levels were determined by qRT-PCR 72 h after *BRD3OS* (*LINC00094*) knockdown. Average of *BRD3OS* (*LINC00094*) knockdown using four different siRNAs targeting *BRD3OS* (*LINC00094*) is shown. (**B**) Levels of MMP-1 and MMP-13 in conditioned media were determined by Western blotting 72 h after *BRD3OS* (*LINC00094*) knockdown. TIMP-1 was used as the loading control. Mean + SD is shown, ** *p* < 0.01; *** *p* < 0.001. Student’s *t*-test.

**Figure 6 cancers-14-03980-f006:**
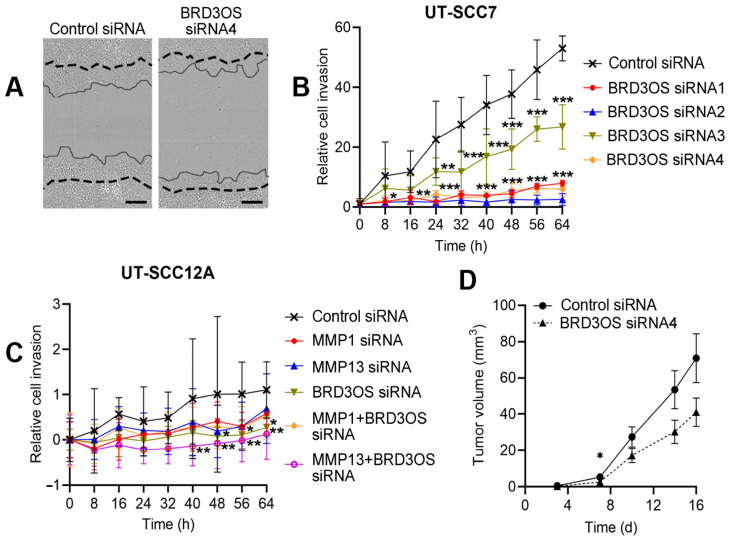
*BRD3OS* (*LINC00094*) knockdown inhibits cSCC cell invasion through collagen I. (**A**) Representative images of invasion assay (64 h) of negative control siRNA and *BRD3OS* siRNA4-treated cSCC cells. (**B**) cSCC cells were transfected with negative control siRNA or four different *BRD3OS* (*LINC00094*) siRNAs, or (**C**) with MMP1 and MMP13 siRNAs in combination with or without *BRD3OS* siRNA siRNA2 and 4 and plated on collagen I 24 h after transfection. Cell monolayer was scratched, and collagen I solution was added in wells and allowed to polymerize, followed by real-time imaging using the IncuCyte S3. Mean ± SD is shown, * *p* < 0.05, ** *p* < 0.01, *** *p* < 0.001, two-way ANOVA, *n* = 4–7. (**D**) cSCC cells (UT-SCC7) were transfected with negative control siRNA (*n* = 8) or *BRD3OS* siRNA4 (*n* = 7) and injected subcutaneously into the back of SCID mice. Tumor growth was measured twice a week. Scale bar = 300 μm. Mean ± SEM is shown. * *p* < 0.05. Student’s *t*-test.

## Data Availability

The RNA-seq data of *BRD3OS* (*LINC00094*) knockdown in cSCC cells is available in the GEO database; accession number GSE205981.

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
