# Peer review of "Super Enhancer-Regulated LINC00094 (SERLOC) Upregulates the Expression of MMP-1 and MMP-13 and Promotes Invasion of Cutaneous Squamous Cell Carcinoma"

_cancers, 2022, doi:10.3390/cancers14163980_

Round 1

Reviewer 1 Report

Manuscript describes LINC00094 possible function in the cutaneous squamous cell carcinoma (cSCC) biology. Authors documented that this special RNA is upregulated in cSCC in comparison to normal keratinocytes. The level of LINC00094 increased with cSCC progression and it influences metalloproteinases expression.

The basic, minor remarks refer:

1. The method for LINC00094 evaluation of tissue samples was a semiquantified method thus what were the positive and negative probes for the basal evaluation and how many scientist estimated histological slides?

2.       As a model for the molecular mechanism examination HACAT cells and cSCC cell lines established from surgically removed cSCC were used.
There is no information about the lines originated from patients although important conclusions have been based on the experimental results with these lines. In the Materials and Methods did not appear even a general name for these lines,  UT-SCC, thus a reader is a little confused when names UT-SCC7, 12, 12A, 59A, 91, 105, 111 appeared in the results. I recommended to give a short description about the criteria for choosing these lines and  what stage of cSCC development they refer to.  

Author Response

Comments and Suggestions for Authors

Manuscript describes LINC00094 possible function in the cutaneous squamous cell carcinoma (cSCC) biology. Authors documented that this special RNA is upregulated in cSCC in comparison to normal keratinocytes. The level of LINC00094 increased with cSCC progression and it influences metalloproteinases expression.

The basic, minor remarks refer:

  1. The method for LINC00094 evaluation of tissue samples was a semiquantified method thus what were the positive and negative probes for the basal evaluation and how many scientist estimated histological slides?

We wish to thank the reviewer for these comments. We have added a Supplementary Figure S2 to illustrate the RNA-ISH with negative (DapB) and positive (PPIB) probes and added text to Results. In addition, the number of observers has been added for the histology evaluation in Methods.

  1. As a model for the molecular mechanism examination HACAT cells and cSCC cell lines established from surgically removed cSCC were used.

There is no information about the lines originated from patients although important conclusions have been based on the experimental results with these lines. In the Materials and Methods did not appear even a general name for these lines,  UT-SCC, thus a reader is a little confused when names UT-SCC7, 12, 12A, 59A, 91, 105, 111 appeared in the results. I recommended to give a short description about the criteria for choosing these lines and what stage of cSCC development they refer to. 

We wish to thank the reviewer for these comments. We have added the names of the cSCC cell lines in Methods and described the cell lines in more detail. We also added more detailed information about HaCaT cell lines to Methods.

Reviewer 2 Report

In the submitted manuscript Piipponen et al. studied super enhancer (SE)-regulated lincRNA BRD3OS in cutaneous squamous cell carcinoma (cSCC) and discovered that higher BRD3OS expression was detected in cSCC cells and their metastases compared to normal skin, actinic keratoses and cSCCs in situ. In addition, RNA-seq analysis after BRD3OS knockdown revealed significantly regulated GO terms cell-matrix adhesion, basement membrane, metalloendopeptidase activity, and KEGG pathways extracellular matrix-receptor interaction. Among the top-regulated genes were MMP1, MMP10 and MMP13. Furthermore, BRD3OS knockdown caused decreased production of MMP-1 and MMP-13 by cSCC cells, suppressed invasion of cSCC cells through collagen I and growth of human cSCC xenografts in vivo.

This manuscript is quite well written, adequate number of proper experiments were conducted, and conclusions were corroborated by results. There are just few minor drawbacks that have to be corrected and further improved:

1) Since currently approved symbol for studied lincRNA is BRD3OS (https://www.genenames.org/data/gene-symbol-report/#!/hgnc_id/HGNC:24742) this symbol should prevail. Therefore, you can eventually write "BRD3OS (BRD3 opposite strand), also named LINC00094". By the way, it was twice misspelled BRDOS3. Also, gene/mRNA/ncRNA symbols were not consistently written in italics throughout the text.

2) Line 52: Replace "temporarily" with "time" because these are not synonyms.

3) Although most methods are self-cited, what is especially problematic for understanding which all cSCC cell lines were used, there should be more details provided so this study would be reproducible:

a) qPCR: State which kit was used for RT; I suppose that RealTimeDesign Software was actually accessed much later than April 17, 2012; provide qPCR cycling conditions; I suppose you used 2^-ddCT method for relative gene expression analysis (https://doi.org/10.1006/meth.2001.1262), standard curve method is usually used for calculating the efficacy of primer pairs and cycling conditions or absolute gene expression analysis.

b) RNA-sequencing: State which kit was used for library preparation; state which read aligner was used; for ALL web-based software provide web address, date when last accessed, and reference; provide references for all used R packages if published in scientific journals; state which fold change and (adjusted) p-value was considered statistically significant for differentially expressed genes.

c) Western blot analysis: Take care that for all used commercial siRNAs and antibodies their catalogue number was provided; provide used dilutions for antibodies; explain how western blot bands were quantified.

d) Lines 190-191: I suppose you used "Incucyte Woundmaker".

e) Statistical analysis: Provide considered level of statistical significance.

4) Explanation why THZ1 and JQ1 were used should come earlier in the text, for instance in 'Methods", not only in 'Results'.

5) In several figure captions you wrote "Mean ± SD is shown" while wast majority of your bar graphs ONLY show mean + SD. It is also unclear why only for Figure 6 "Mean ± SEM" presentation was used?

Author Response

Comments and Suggestions for Authors

In the submitted manuscript Piipponen et al. studied super enhancer (SE)-regulated lincRNA BRD3OS in cutaneous squamous cell carcinoma (cSCC) and discovered that higher BRD3OS expression was detected in cSCC cells and their metastases compared to normal skin, actinic keratoses and cSCCs in situ. In addition, RNA-seq analysis after BRD3OS knockdown revealed significantly regulated GO terms cell-matrix adhesion, basement membrane, metalloendopeptidase activity, and KEGG pathways extracellular matrix-receptor interaction. Among the top-regulated genes were MMP1, MMP10 and MMP13. Furthermore, BRD3OS knockdown caused decreased production of MMP-1 and MMP-13 by cSCC cells, suppressed invasion of cSCC cells through collagen I and growth of human cSCC xenografts in vivo.

This manuscript is quite well written, adequate number of proper experiments were conducted, and conclusions were corroborated by results. There are just few minor drawbacks that have to be corrected and further improved:

We wish to thank the reviewer for these encouraging comments.

1) Since currently approved symbol for studied lincRNA is BRD3OS (https://www.genenames.org/data/gene-symbol-report/#!/hgnc_id/HGNC:24742) this symbol should prevail. Therefore, you can eventually write "BRD3OS (BRD3 opposite strand), also named LINC00094". By the way, it was twice misspelled BRDOS3. Also, gene/mRNA/ncRNA symbols were not consistently written in italics throughout the text.

We wish to thank the reviewer for these comments. We have changed the naming in the Abstract and Introduction as suggested. We have also replaced LINC00094 as BRD3OS (LINC00094) in the text and in Figures, Supplementary Figures and Tables. We also went carefully through the text and corrected the naming of BRD3OS and the gene, mRNA and ncRNA names as italics.

2) Line 52: Replace "temporarily" with "time" because these are not synonyms.

We wish to thank the reviewer for these comments. This typo was corrected as “temporally” in the text.

3) Although most methods are self-cited, what is especially problematic for understanding which all cSCC cell lines were used, there should be more details provided so this study would be reproducible:

  1. a) qPCR: State which kit was used for RT; I suppose that RealTimeDesign Software was actually accessed much later than April 17, 2012; provide qPCR cycling conditions; I suppose you used 2^-ddCT method for relative gene expression analysis (https://doi.org/10.1006/meth.2001.1262), standard curve method is usually used for calculating the efficacy of primer pairs and cycling conditions or absolute gene expression analysis.

We wish to thank the reviewer for these comments. We have added detailed information about RT method and the qPCR cycling conditions in Methods.

Last accession date of RealTimeDesign Software was removed.

We are using standard curve method for gene expression analysis in qPCR. We prepare the dilutions for standard curve so that we combine cDNA from all samples and then dilute it to four concentrations. We add standard samples for all primer/probe sets on every analyzed plates/runs. This provides us a routine validation for methodology.

  1. b) RNA-sequencing: State which kit was used for library preparation; state which read aligner was used; for ALL web-based software provide web address, date when last accessed, and reference; provide references for all used R packages if published in scientific journals; state which fold change and (adjusted) p-value was considered statistically significant for differentially expressed genes.

We explained RNAseq method in more detail and added the filtering criteria to Methods. We also added a reference for software and a web page for online tool in Methods. 

  1. c) Western blot analysis: Take care that for all used commercial siRNAs and antibodies their catalogue number was provided; provide used dilutions for antibodies; explain how western blot bands were quantified.

We have added the missing catalogue numbers to Supplementary Table S2  and the antibody dilutions in the Methods. We also explain how the Western blot bands are quantitated in Methods.

  1. d) Lines 190-191: I suppose you used "Incucyte Woundmaker".

This is now corrected in the text.

  1. e) Statistical analysis: Provide considered level of statistical significance.

We have added the threshold value considered statistically significant to Methods.

4) Explanation why THZ1 and JQ1 were used should come earlier in the text, for instance in 'Methods", not only in 'Results'.

We wish to thank the reviewer for this comment. We have added the description of these inhibitors in Methods.

5) In several figure captions you wrote "Mean ± SD is shown" while wast majority of your bar graphs ONLY show mean + SD. It is also unclear why only for Figure 6 "Mean ± SEM" presentation was used?

We wish to thank the reviewer for these comments. We changed SD in Figure legends as mentioned by the reviewer.

In our results SD is used to express the variability of the data from mean for exactly measurable values as descriptive statistics. We have used SEM only for Figure 6D showing the xenograft tumor volumes. The volumes were calculated using the formula (V = (length × width2)/2), and these parameters for each tumor were determined by manual measurement using caliper, and therefore are estimates rather than exact values, because the xenograft tumor model bears inherent variability. By using SEM for this experiment we evaluate and show how well the model truly represents the reality and entire population